# Mismatch Negativity and Stimulus-Preceding Negativity in Paradigms of Increasing Auditory Complexity: A Possible Role in Predictive Coding

**DOI:** 10.3390/e23030346

**Published:** 2021-03-15

**Authors:** Francisco J. Ruiz-Martínez, Antonio Arjona, Carlos M. Gómez

**Affiliations:** Human Psychobiology Lab, Experimental Psychology Department, University of Sevilla, 41018 Seville, Spain; aarjona@us.es (A.A.); cgomez@us.es (C.M.G.)

**Keywords:** mismatch negativity, stimulus preceding negativity, contingent negative variation, predictive coding, regularity-violation hypothesis, intertrial interval

## Abstract

The auditory mismatch negativity (MMN) has been considered a preattentive index of auditory processing and/or a signature of prediction error computation. This study tries to demonstrate the presence of an MMN to deviant trials included in complex auditory stimuli sequences, and its possible relationship to predictive coding. Additionally, the transfer of information between trials is expected to be represented by stimulus-preceding negativity (SPN), which would possibly fit the predictive coding framework. To accomplish these objectives, the EEG of 31 subjects was recorded during an auditory paradigm in which trials composed of stimulus sequences with increasing or decreasing frequencies were intermingled with deviant trials presenting an unexpected ending. Our results showed the presence of an MMN in response to deviant trials. An SPN appeared during the intertrial interval and its amplitude was reduced in response to deviant trials. The presence of an MMN in complex sequences of sounds and the generation of an SPN component, with different amplitudes in deviant and standard trials, would support the predictive coding framework.

## 1. Introduction

Our brain processes perceptual information to create an adaptive response to the environment. Predictive mechanisms are hypothesized to minimize the difference between internal representations (generated according to the environmental cues) and sensory inputs [1]. Predictive coding assumes that the brain implements a hierarchical generative model of its sensory inputs, which it optimizes by minimizing prediction errors between top-down predictions and bottom-up streams of sensory inputs. Each new stimulus perception creates a loop between the sensation and the internal model to minimize its discrepancy. The predictive error would cause an increase in neural activity, whose objective is to update the model and improve its compatibility with the environment [2,3,4,5]. This approach is formally expressed in the so-called Bayesian Brain Hypothesis [2], and it has been applied to different fields as sensory processing [6,7], sensory-motor activity [8,9,10], language [11], or attention [12].

The auditory mismatch negativity (MMN) is induced by the repetition of an auditory stimulus pattern (standard) unexpectedly replaced by a different pattern (deviant) [13]. This deflection is considered as a preattentional index of deviant detection which compares the extracted rules from the previous stimulus history with the current stimulation pattern, without requiring to pay attention voluntarily or execute a behavioral response [14]. It can be induced by differences in amplitude, frequency, duration, and pitch features of the deviant/standard stimuli, and it is characterized by a fronto-central negativity between 100 and 250 ms after the deviant stimulus onset [15,16]. It has been suggested that the psychophysiological phenomenology of MMN could be explained by a basic habituation process [17,18,19]. This proposal has been challenged by the so-called regularity-violation hypothesis [20], a predictive model theory of the MMN which establishes that the brain extracts regularities from the environment permitting it to compute probabilities of future auditory events. This theory is largely based on the fact that MMN appears not only to single features but also to complex organizations of sounds, reflecting that the subjects are extracting complex rules of sounds patterns and signaling the presence of the deviant patterns [21,22,23].

Inspired by the predictive coding framework for MMN, a neurophysiologically based computational model of MMN has been proposed [24]. The architecture of the model includes four interconnected layers: (i) a thalamic layer where the standard and deviant features are represented, and then a series of cortical layers; (ii) a predictive error layer which integrates the activity of the thalamic layer with the predictive input of the predictive layer (layer iii), which receives the inputs from a memory layer (layer iv) of recent inputs. Interestingly, the prediction error layer would receive excitatory inputs from the thalamic layer, and a tonic inhibitory input from the predictive layer. If the predictive tonic inhibition is not able to cancel the phasic excitatory input from the thalamic layer, then a prediction error signal would be produced in the prediction error layer, which would be indexed by the MMN. Although the model has been successfully tested for the analysis of single deviant features, the authors propose that the model would be able to accommodate the prediction errors generated by complex features as melodic contours, by adding a new hierarchically superior layer of neurons sensitive to these complex features. The model suggests that the complex predictive signals should feed back to the prediction error layer.

A possible neurophysiological candidate for this possible predictive signal is the stimulus-preceding negativity (SPN) [25]. The SPN is a negative component that appears after an action whose outcome must be assessed by a stimulus to be delivered after a few seconds [26]. This component is elicited before a sensory stimulus and does not require an immediate motor response; it can be considered as a sensory attentional anticipation signal not contaminated from motor response preparation. Although the SPN was initially discovered in time estimation tasks, this negativity appears more clearly in experimental paradigms with an affective or motivational component [27]. The insular cortex has been proposed as a major source for the SPN [28].

A slow wave related to the SPN is the contingent negative variation (CNV) [29,30,31]. CNV is a fronto-central negative slow wave, starting around 400 ms after a warning or cue signal, which is usually associated with sensory and motor preparation [3,25,32], reflecting the expectation generated by a warning cue (S1) about the appearance of the target stimulus (S2) and the intended response [29,30,31]. The CNV has been shown to be related to the expectancy of the next target as a function of the outcomes of previous trials [10]. If the target stimulus following a CNV coincides with the location indicated by the cue inducing the CNV, the amplitude of CNV in the next trial presented a higher amplitude compared to if the target was presented in an unexpected location. Therefore, it is possible that SPN would behave in a similar manner, increasing amplitude after a standard trial and decreasing amplitude after an unexpected deviant.

Based on previous literature, the present report explores whether the information about the deviant stimulus ending of each trial is reflected by the MMN and if an SPN component is present in the intertrial interval. The aims would be (i) to test if the MMN appears in trials with deviant endings of sequences of predictable increasing (or decreasing) auditory frequencies. The trial sequences would be constituted by frequencies that are repeated frequently (first-order complexity) or changing frequently throughout the experiment (second-order complexity). With this experimental paradigm it would be possible to test the sensitivity of MMN to frequency contour rules of deviants of different complexity; (ii) the predictive coding signal would be analyzed through the possible presence of an SPN after MMN, which would indicate the strength of the prediction.

Due to the well-known frontal and central topography of the MMN which has been broadly described [14,33], only fronto-central electrodes in early latencies would be analyzed for the MMN. Whereas for the analysis of the SPN, the whole scalp and intertrial time window post-MMN would be analyzed with an exploratory approach. All the analyses would be computed using the cluster mass permutation test which offers the possibility to explore the whole spatiotemporal distribution of event-related potentials (ERPs) [34,35].

## 2. Materials and Methods

### 2.1. Participants

A sample of 31 healthy subjects (15 males and 16 females) between 21 and 30 years old were recorded in both experimental conditions. One female subject was excluded from the analysis of the first-order complexity due to problems during the EEG recording. Thus, this experimental condition included a sample of 30 subjects (15 males and 15 females) between 21 and 30 years old (mean = 26.55), whereas the second-order complexity included a sample of 31 subjects (15 males and 16 females) between 22 and 29 years old (mean = 27.11). The subjects were recruited from a middle socio-economic background and they did not report any neurological disease or psychological impairments. Experiments were conducted with the informed and written consent of each participant, following the Helsinki Protocol. The study was approved by the Bioethical Committee of the Junta de Andalucía.

### 2.2. Stimuli

Auditory stimuli were presented through two speakers (Dell, model A215), positioned on either side of the participant’s head at a comfortable listening volume of 65 dB (measured with a Velleman-DVM1326 sound-meter). Every tone was a sinusoidal wave with a duration of 200 ms including 20 ms rise and fall time, created using an online tone generator (https://www.wavtones.com/functiongenerator.php, last accessed on 15 April 2019) and 2.3.2 of Audacity (R) recording and editing software. The experiment was programmed using E-Prime version 2.0.

Two different conditions were presented: first-order and second-order complexity conditions composed of a sequence of four stimuli for each trial (Figure 1). Increasing and decreasing frequencies were counterbalanced between subjects to control that the obtained results were not a by-product of the influence of the direction of activation in the auditory cortex.

Within the first-order complexity, the standard trial presented the following four stimuli in the ascending frequency sequence: 1300 Hz, 1600 Hz, 1900 Hz, 2200 Hz; the deviant trial was: 1300 Hz, 1600 Hz, 1900 Hz, 1000 Hz. In the descending frequency sequence, the standard trial presented the following stimuli: 1900 Hz, 1600 Hz, 1300 Hz, 1000 Hz, while the deviant trial was: 1900 Hz, 1600 Hz, 1300 Hz, 2200 Hz.

For the second-order complexity, within the frequency increase sequences, there were five standard trials: “800 Hz, 1100 Hz, 1400 Hz, 1700 Hz”; “1300 Hz, 1600 Hz, 1900 Hz, 2200 Hz”; “1800 Hz, 2100 Hz, 2400 Hz, 2700 Hz”; “2300 Hz, 2600 Hz, 2900 Hz, 3200 Hz”; “2800 Hz, 3100 Hz, 3400 Hz, 3700 Hz”. Whereas the deviant trials were “800 Hz, 1100 Hz, 1400 Hz, 500 Hz”; “1300 Hz, 1600 Hz, 1900 Hz, 1000 Hz”; “1800 Hz, 2100 Hz, 2400 Hz, 1500 Hz”; “2300 Hz, 2600 Hz, 2900 Hz, 2000 Hz”; “2800 Hz, 3100 Hz, 3400 Hz, 2500 Hz”. For the decreasing frequency sequences, the standard trials were “1400 Hz, 1100 Hz, 800 Hz, 500 Hz”; “1900 Hz, 1600 Hz, 1300 Hz, 1000 Hz”; “2400 Hz, 2100 Hz, 1800 Hz, 1500 Hz”; “2900 Hz, 2600 Hz, 2300 Hz, 2000 Hz”; “3400 Hz, 3100 Hz, 2800 Hz, 2500 Hz”. Whereas the deviant trials were: “1400 Hz, 1100 Hz, 800 Hz, 1700 Hz”; “1900 Hz, 1600 Hz, 1300 Hz, 2200 Hz”; “2400 Hz, 2100 Hz, 1800 Hz, 2700 Hz”; “2900 Hz, 2600 Hz, 2300 Hz, 3200 Hz”; “3400 Hz, 3100 Hz, 2800 Hz, 3700 Hz”.

Every single stimulus had a 200 ms duration, the interstimulus interval was 100 ms and the intertrial interval lasted 1900 ms. The total duration of each stimulus sequence in one trial was 1100 ms. The total trial duration was 3 s. An example of the experimental conditions is displayed in Figure 1.

Each experimental condition had 241 standards and 61 deviant trials, a total of 604 trials per subject. The total recording time per subject was 30 min and 20 s. The subjects watched a silent movie during the recording session.

With the objective to perform future studies analyzing trial dyads in change (deviant-standard and standard-deviant, coded as DS and SD, respectively) or no-change (standard-standard, deviant-deviant, coded as SS and DD, respectively) conditions, we arranged the order presentation to obtain a sufficient number of the condition deviant-deviant that if arranged randomly would present a very scarce representation.

For this reason, the experiment was organized in 59.1% SS dyads, 15.7% DS dyads, 15.7% SD dyads, and 9.5% DD dyads. To maintain the effect of the standard during the full experiment, the dyads were distributed in a pseudo-randomized way so that after every dyad with a deviant trial, at least one SS trial dyad appears.

### 2.3. EEG Recordings

The recordings were performed in an acoustically isolated room. The participants sat in a comfortable chair positioned between the speakers, facing a laptop in which they watched a silent movie while the experimental session was released. Subjects were asked to stay calm and look at the screen, while minimizing blinks and face and head movements. The recordings were performed at different times of the day between 10 a.m. and 7 p.m., without receiving any information about previous sleep.

EEG was recorded from 20 electrodes aligned with the 10–20 system: (Fp1, Fp2, F7, F3, Fz, F4, F8, T3, C3, Cz, C4, T4, T5, P3, Pz, P4, T6, O1, O2) including a ground electrode. Ocular and mastoids electrodes were avoided to facilitate a future replication of this protocol in studies with clinical populations of different ages, taking into account, that their installation involves an increase in time and it can be aversive for participants with somatosensory reactivity such as autism spectrum disorder subjects. For this reason, the signal amplitudes in the electrodes Fp1, Fp2, F7, and F8 were used to monitor blink artifacts and ocular movements.

EEG data were recorded unfiltered and an average reference was used. Impedance was maintained below 10 KOhms throughout the recording time. Data were recorded in direct current mode at 1024 Hz, with a 20,000-amplification gain using a commercial analog–digital acquisition and analysis board (ANT). Data were filtered offline.

### 2.4. Data Analysis

EEG recordings were analyzed with the EEGLAB version 14.1.1 and Matlab 2016a software packages. To correct the EEG for blinks, ocular movements, and muscle artifacts, an Independent Component Analysis (ICA) was used. These components were removed, and the EEG signal was reconstructed. The mean of removed components in the first-order complexity condition was: 3.9 ± 1.2 SD, range: 2–7; and for the second-order complexity condition: 4 ± 1.3 SD, range: 2–8. EEG recordings were high-pass (0.05 Hz cut-off) and low-pass filtered (45 Hz cut-off).

ERPs were obtained only for the last stimulus of the stimulus sequence composing a trial, given our interest in comparing deviants with standards, and that deviants were only present in the fourth position of the stimulus train.

With a descriptive purpose the average of the ERPs obtained for all the subjects, for the standard and deviant trial, in both experimental conditions (first and second-order complexity) was plotted in Figure 2. The time window presented was −100 to 3200 ms in order to show the full trial (0 to 3000 ms), the baseline used (−100 to 0 ms, before the first stimulus of the stimuli sequence, S1), and the beginning of a new trial (3000 to 3200 ms). The electrode selected was Fz because it is the most related with the MMN generation [14].

Epochs were segmented only for the last stimulus, in time windows of −100 to 1900 ms. All the epochs in which the EEG exceeded ±70 microvolts in any channel were rejected for analysis. The mean of accepted trials in the first-order complexity standard trials was: 236.5 ± 9.69 SD, range: 192–241; and 60.07 ± 1.55 SD, range: 55–61 for the deviant trials. Whilst in the second-order complexity 237.26 ± 6.56 SD (range: 211–241) trials were accepted for standard and 59.68 ± 2.59 SD (range: 48–61) for the deviant trials. Epochs were averaged using ERPLAB, resulting in two different averages depending on the type of trial (standard or deviant) for each participant, for both experimental conditions (first and second-order complexity). Standard and deviant trials were compared in both paradigms for all the subjects, using the STUDY function of EEGLAB. With this objective, the data were analyzed with a paired statistic using a field trip toolbox implemented in EEGLAB for computing the cluster mass permutation [36] with an alpha level of *p* ≤ 0.05 and computing 1000 randomizations.

Two time windows were analyzed depending on the ERP component of interest. For the MMN the analyzed time window was of −50 ms pre-stimulus to 300 ms, because we expected to find the MMN in a time window between 150 and 250 ms. Another time window analyzed was between −100 (pre-stimulus) and 2000 ms to study the SPN late ERP, which should extend its presence in a broad time window. The number of analyzed electrodes was different in each time window. For the MMN, the analyzed electrodes were: F3, Fz, F4, C3, Cz, C4, because the MMN response is recorded in the frontal and central areas [14,33]. The inversion polarity in temporal areas was not analyzed to reduce the number of multiple comparisons. Whereas for the SPN analyses all the scalp electrodes were analyzed in an exploratory manner to observe the possible localization of an anticipatory process without any a priori preconception.

The topographies of the grand average in the MMN latency (100–250 ms) of the ERPs, induced by standards, deviants, and the results of subtracting the standards from the deviants (MMN) were displayed. For displaying the topography of the SPN grand average of standard, deviant, and the difference wave obtained of subtracting the standards from the deviants, the time window of 400–2000 ms was used. This procedure was applied to obtain the voltage topographies for the first and second-order complexity conditions.

## 3. Results

Figure 3 and Figure 4 show the results obtained in the first-order and second-order complexity conditions for the comparisons between standard and deviants in the selected time window (from −50 to 300 ms, electrodes, F3, Fz, F4, C3, Cz, C4). The gray shaded areas indicate the latencies in which exists a significant difference (*p* ≤ 0.05) between standards and deviants.

As can be observed in Figure 5 and Figure 6, the results obtained in the analysis of the SPN show significant differences between standard and deviants due to a higher amplitude of the SPN during standard trials in both experimental conditions. The latencies in which the differences are statistically significant presented a higher duration in the comparisons of the first-order complexity regarding the second-order complexity. The scalp topography showed a reverse polarity with respect to the scalp topography of the MMN component with positivity in fronto-central electrodes and negativity in temporal and temporo-parieto-occipital electrodes.

## 4. Discussion

The MMN has shown to be sensitive to deviant trials for the first-order complexity (repeated frequencies) and second-order complexity (variable frequencies from trial to trial) suggesting that habituation would not be the cause for MMN. The latter result and the presence of an SPN in the intertrial interval, which presents a higher amplitude in the standard with respect to the deviant trial, suggest that MMN would be a neurophysiological adaptive index of prediction error, while the SPN would be indexing the stimulus expectancy for the next trial.

### 4.1. Mismatch Negativity

The ERPs for the first-order and second-order complexity showed higher negativity in deviants than in standard during the MMN time window (around 100–250 ms). The subtraction of the standard from the deviant showed a negative fronto-central topography. These results suggest that an MMN component similar to the odd-ball MMN has been obtained in the present experiment [14,37,38,39].

The MMN obtained in the second-order complexity supports other studies that have also used complex tones sequences to elicit MMN, proving that this component can also be induced by abstract rules, and not only by the difference in the physical features of the compared stimuli [24,40,41]. In the second-order complexity condition, the difference in frequency between the tones between successive trials would work as the abstract rule being implemented in the network generating the MMN because of two reasons. First of all, this rule is present in every trial; secondly, the low repetition of the same tones in the stimuli prevent making a prediction exclusively based on the physical features, and then some sort of inference about the physical characteristic of the next stimulus must be generated to obtain an MMN likeness in the deviant trials. Therefore, what would be learned is the abstract rule of decreasing or increasing auditory frequencies, and then the MMN would be elicited by the deviant violating the frequency directionality (increasing or decreasing, depending on the particular rule implementation). In the end, the auditory system would probably generate an expectation for a given frequency, but this would be generated by an abstract rule following the regularity-violation interpretation of the MMN [20], and using a generative model in the predictive coding terminology [1]. 

The present results suggest that MMN is generated by predictive coding instead of being generated by physiological mechanisms related to the adaptation hypothesis [42]. To justify this point, it is necessary to describe the physiological organization of the primary auditory cortex and its relation to the adaptation hypothesis for MMN generation. The primary auditory cortex is organized in microcolumns arranged in an ordered way depending on the tone frequency analyzed, shaping a tonotopic mapping which has been verified with MEG [43,44], EEG [45], blood flow [46], and intracranial microelectrodes [47].

The adaptation hypothesis for MMN generation would consider this component as the consequence of the difference between the neural attenuation caused by the repetition of similar stimuli during standards presentations, and the non-habituated activation produced by a different stimulus during deviant presentations [17,18,19,48]. Therefore, the adaptation hypothesis could be refused because the very large variability of tones released in the complex paradigm, in both deviant and standard trials, should stimulate the same population of neurons a few times (due to the tonotopic organization of the primary auditory cortex described above), so the habituation of the standards should be very low. On the other hand, the possibility of MMN being generated by predictive coding does not make any assumption about the number of times a given stimulus is presented but relies on the comparison between the predicted and the current stimulus [24].

Based on the similarity observed in the deviant topographies of both complexity conditions, our study would support the proposals which suggest that the auditory cortex processes not only the sensory information but also other categorical parameters involved in the formation of prediction based on more abstract rules, instantiated in the present report by the frequency directionality of the tones [23,49].

The MMN can be followed by a P3a reflecting the involuntary attention capture [14] and then continued by the so-called Response Orientation Negativity (RON) [50]. These components were not clearly defined in the present experiment in which attention was directed to a mute movie and the auditory stimulation was completely irrelevant. The irrelevance of auditory stimulation and the need of creating an abstract rule for generating deviant-related signals would be the cause of the limited processing of auditory deviation obtained in the present report [50,51,52]. Although certainly, this aspect would need further research.

In a different type of paradigm, the visuo-auditory central cue Posner paradigm, an increase in the P300 component to invalidly cued targets has been shown [53,54]. These results were interpreted as if the P300 was related to the updating of the conditional probability between cues and targets. More formally, the P300 was related to the surprise parameter [10] extracted from a Bayesian hierarchical model [55]. Therefore, the updating of predictive models would be indexed by different ERP components, i.e., MMN in passive tasks and P300 in active tasks, at different hierarchical levels. The results suggest that updating of internal models would be a general property of brain systems [1].

### 4.2. SPN Component

An important point that has not been sufficiently addressed in the MMN literature is how information about the trial history is transferred to the next trials. This would be necessary to provide information about the current predictive model to the prediction error layer. In the present report, an SPN has been recorded in both complexity conditions on the whole intertrial interval. The term SPN to define the slow potential obtained in the present report has been chosen to term this component because it shares with SPN the characteristic of being a slow potential induced before a sensory stimulus which does not require a response. However, it is different from classical SPN in topography and right hemisphere preponderance, given that SPN to expected auditory stimuli is characterized by negative voltage in frontal areas [56], while in the present report the negativity appears in posterior sites, accompanied by positive voltage in anterior sites, displaying a mirror inverse topography to the MMN. Moreover, the motivational and affective aspects [27] that modulate the SPN are not tested in this experiment, where a passive experimental condition is applied.

A similar anticipatory ERP, also elicited by a passive auditory condition, is described in the study of Bianco et al. 2020 [57]. This slow potential, which they named as auditory Positivity or aP, is also located in frontal areas and precedes the following stimulus (emerges at −800 ms before stimulus onset in frontal sites). Additionally, they also observed that the post-stimulus component (N1) showed a similar distribution with an opposite polarity to the aP, in a similar manner as it is found in the present report when topographies in Figure 3 and Figure 5, and in Figure 4 and Figure 6 are compared.

The similitudes found between the present SPN and the aP, and the conclusions proposed by the authors of the study about the performance of the auditory cortex in a predictive passive task [57], support the hypothesis explained before about the role of the auditory SPN in predictive coding. A role particularly supported by the SPN modulation by the trial outcome, as obtained in the present report.

As far as we know, no previous study has shown such a type of SPN after MMN. The regularity-violation hypothesis proposes that MMN is reflecting the prediction error if there is an inadequacy between the predicted sound and the current sound [20]. In the same sense, quantitative neural modeling of the MMN system [24] proposed that the MMN would be computed as the difference between an excitatory thalamic input with an inhibitory tonic predictive signal, if both signals present a similar amplitude no MMN is produced, however, if an amplitude difference exists an MMN is generated in the predictive error layer, that would be recorded as MMN from the scalp recordings. Therefore, the model would predict a similar topography for MMN and the predictive signal, and although it is difficult to predict the polarity generated in the scalp surface from the microcolumns processing, an opposite polarity between both signals would be highly suggestive of opposite synaptic activation and inhibition in the prediction error layer as proposed in the [24] MMN model. The presence of the SPN during the intertrial interval with similar topography and inverted polarity with respect to MMN suggests that the SPN would represent the predictive signal. In this sense, the similarity in topographies has been proved to be a strong indicator of similar neural sources [58]. Therefore, the inverse polarity and similar topography of MMN and SPN would be interpreted as suggesting an opposite action of the thalamic and predictive signals in similar areas to those generating the MMN, as predicted in the MMN computational model of [24].

The SPN has been shown to be modulated by the belief entropy parameter [59], linking this component to the predictive coding of the next expected stimuli and represents a candidate for sustaining the predictive coding for future stimuli in MMN-type experiments. The adaptive characteristic of SPN, presenting a higher amplitude after a standard than after a deviant also suggests a dynamical change in a trial-by-trial basis of the predictive signal, similarly to how it occurs in the CNV in the central cue Posner paradigm [10,31]. In this sense, the CNV has been formally related to the expectancy parameter by correlating EEG during the CNV period with the expectancy parameter obtained from a hierarchical Bayesian model [10,55].

The SPN and the MMN were generated without asking any particular task of subjects that should be relatively unaware of the auditory stimulation characteristics. The MMN has been proposed to index preattentive auditory processing [14]. For the SPN, the computational relationship with predictive coding of the next stimulus can be suggested.

## 5. Conclusions

The present study suggests that there is a continuous updating of the neurophysiological signals indexing the processing of sequences of sounds. The extracted rule would operate at different levels of abstraction. This behavior was obvious for MMN, which would implement prediction error, but also for the SPN, which would transfer an anticipatory prediction for the next trial.

## Figures and Tables

**Figure 1 entropy-23-00346-f001:**
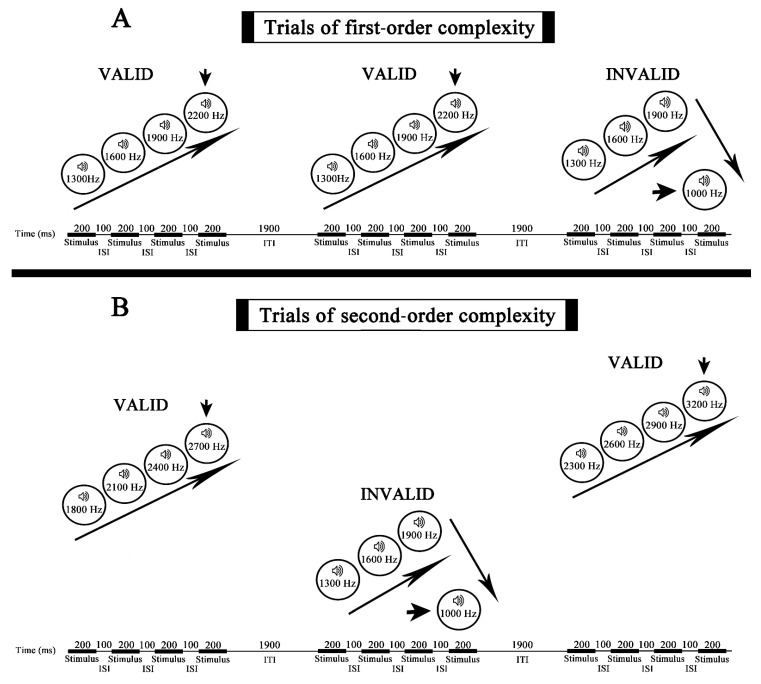
Experimental paradigm. (**A**) The figure shows the order and features (frequency and duration) of every stimulus in the standard and deviant trials for the first-order complexity condition. The arrow pointing to the last sound of each trial indicates that all the analyses are performed using this stimulus as a trigger for averaging (top). The durations of interstimulus intervals and intertrial intervals are also displayed. (**B**) The same for the second-order complexity condition (bottom).

**Figure 2 entropy-23-00346-f002:**
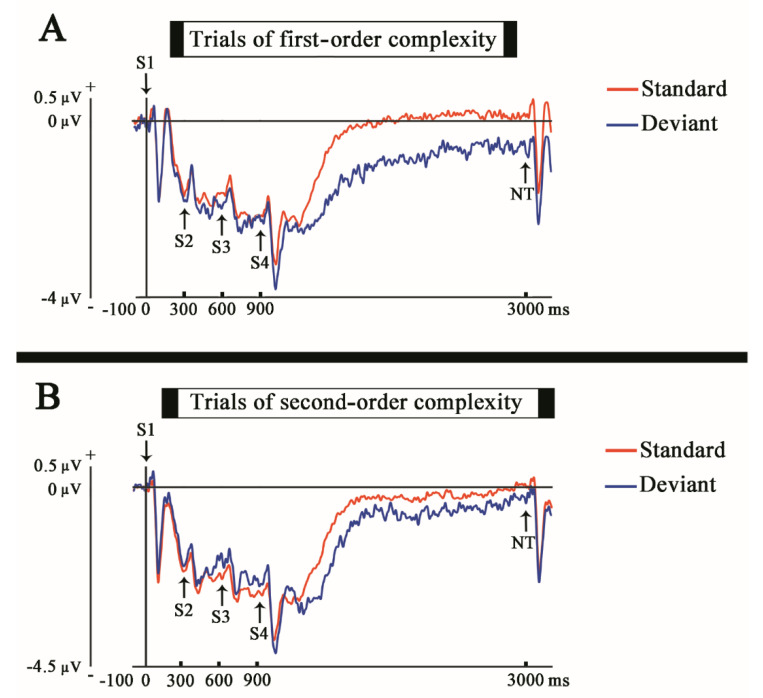
ERPs of standard and deviant in both experimental conditions. (**A**) The ERPs show the voltage grand average of every standard and deviant trial, in the electrode Fz, for the first-order complexity condition. The arrows pointing to the potentials indicate the onset of every stimulus according to their order of apparition (Sx) and the onset of the next trial (NT) (top). (**B**) The same for the second-order complexity condition (bottom).

**Figure 3 entropy-23-00346-f003:**
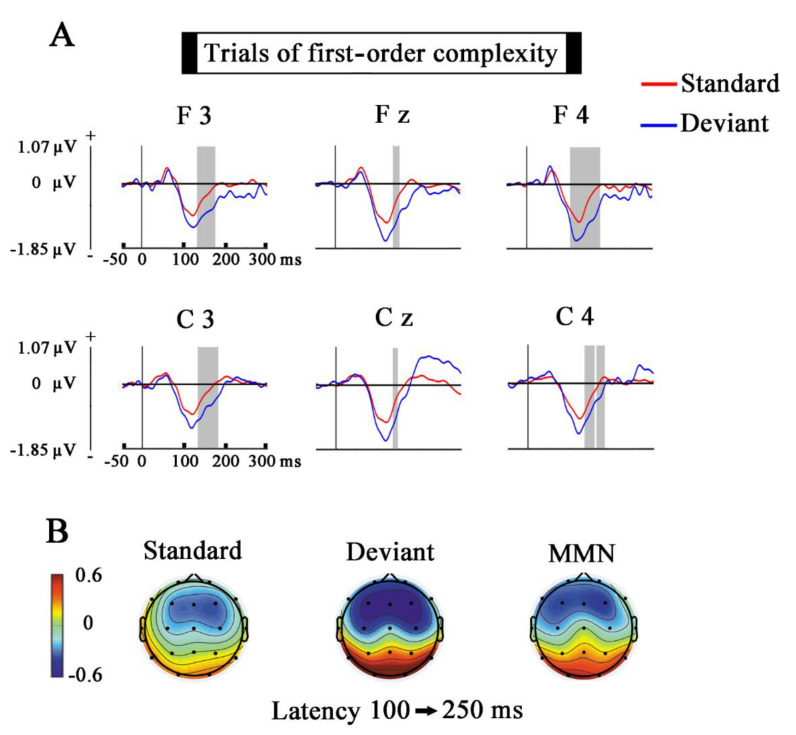
Comparison of the Standard and Deviant trials (first-order complexity). (**A**) The ERPs show the voltage grand average for the last stimulus of every standard and deviant trials. Shaded gray areas correspond to the time windows in which significant differences (*p* ≤ 0.05) were obtained with the cluster mass permutation statistics. Notice that significant differences appear in the mismatch negativity (MMN) latency. (**B**) Topographies of standard, deviant, and MMN in the latency window of 100–250 ms. Notice the fronto-central topography of the MMN.

**Figure 4 entropy-23-00346-f004:**
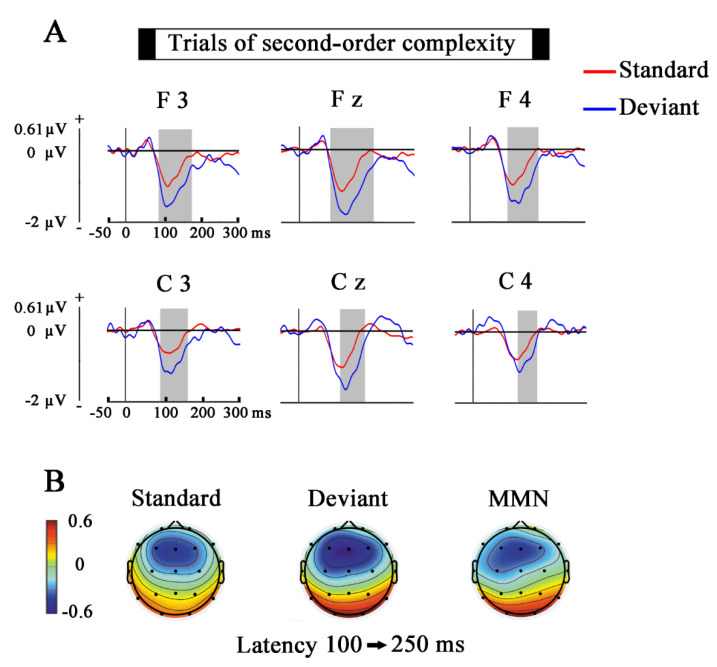
Comparison of the Standard and Deviant trials (second-order complexity). (**A**) The ERPs show the voltage grand average for the last stimulus of every standard and deviant trials. Shaded gray areas correspond to the time windows in which significant differences (*p* ≤ 0.05) were obtained with the cluster mass permutation statistics. Notice that significant differences appear in the MMN latency. (**B**) Topographies of standard, deviant, and MMN in the latency window of 100–250 ms. Notice the fronto-central topography of the MMN.

**Figure 5 entropy-23-00346-f005:**
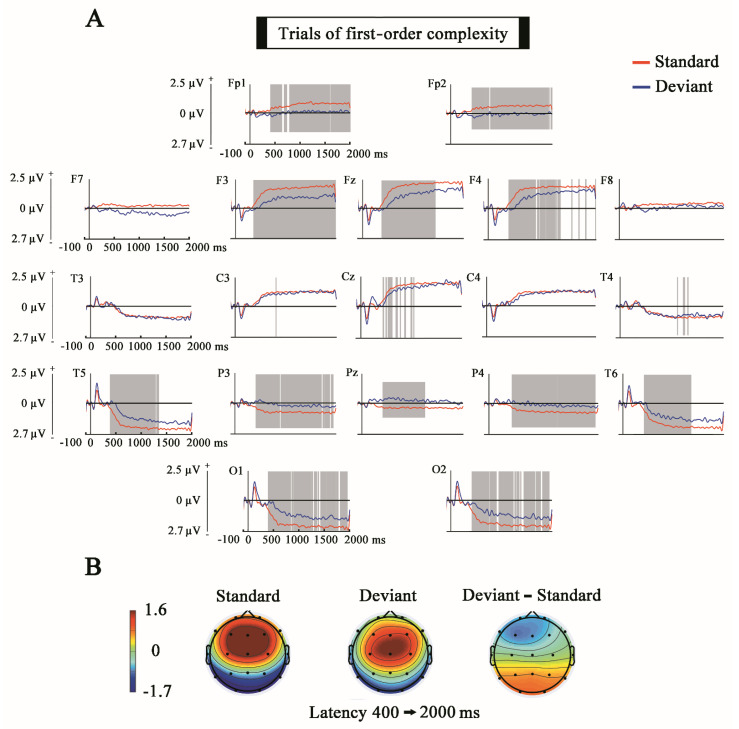
Comparison of the Standard and Deviant trials for the full epoch (first-order complexity). (**A**) The ERPs show the voltage grand average for the last stimulus of every standard and deviant trials. Shaded gray areas correspond to the time windows in which significant differences (*p* ≤ 0.05) were obtained with the cluster mass permutation statistics computed for the whole time window. Notice that significant differences appear in the stimulus-preceding negativity (SPN) latency. (**B**) Topographies of standard, deviant and difference wave in the latency window of 400–2000 ms. Notice the positive fronto-central topography of the SPN and the negative frontal topography of the difference wave.

**Figure 6 entropy-23-00346-f006:**
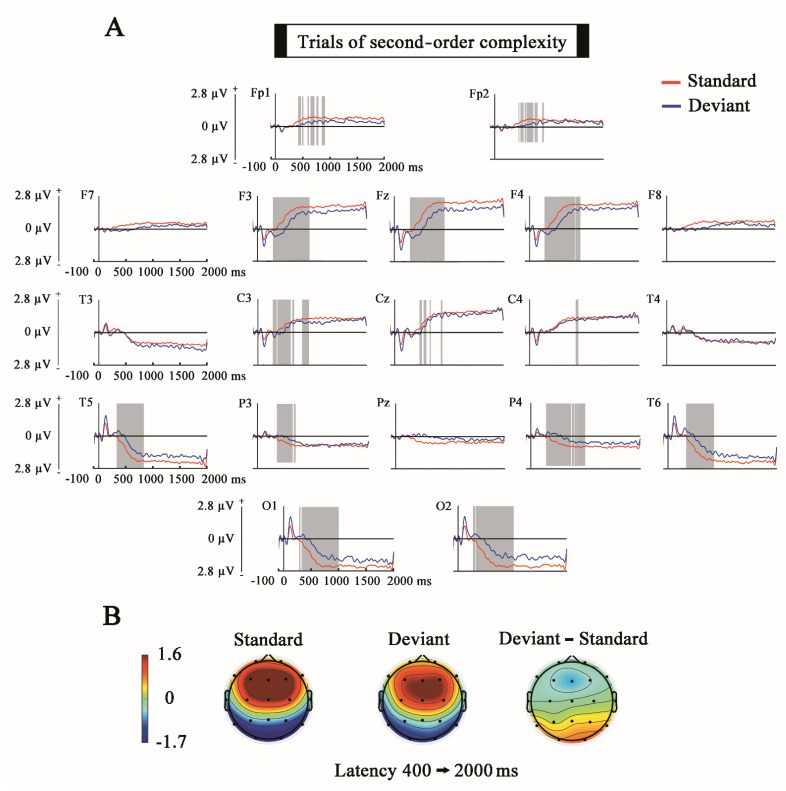
Comparison of the Standard and Deviant trials for the full epoch (second-order complexity). (**A**) The ERPs show the voltage grand average for the last stimulus of every standard and deviant trials. Shaded gray areas correspond to the time windows in which significant differences (*p* ≤ 0.05) were obtained with the cluster mass permutation statistics computed for the whole time window. Notice that significant differences appear in the SPN latency. (**B**) Topographies of standard, deviant and difference wave in the latency window of 400–2000 ms. Notice the positive fronto-central topography of the SPN and the negative frontal topography of the difference wave.

## Data Availability

The data presented in this study are available on request from the corresponding author.

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
