# Peer review of "Mismatch Negativity and Stimulus-Preceding Negativity in Paradigms of Increasing Auditory Complexity: A Possible Role in Predictive Coding"

_entropy, 2021, doi:10.3390/e23030346_

Round 1

Reviewer 1 Report

The authors show data from an elegant EEG experiment, displaying mismatch negativity and stimulus preceding negativity during the passive listening to simple and more complex acoustic sequences.

Some of the sentences in this manuscript are hard to understand, supposedly due to a non-optimal choice of words and/or prepositions. However, the introduction and discussion sections are sound and understandable.

The methods and results sections are very clear.

Concrete comments and suggestions:

1.) Abstract, first line: "... has been considered a marker of preattentive index of ..." -> I think it should be "... has been considered a preattentive index of ..."

2.) Although indirect and direct evidence for predictive coding as a possible implementation of Bayesian computations in the brain is mounting, I would still formulate the first sentence of the introduction a bit more carefully (Line 26-27): "Predictive mechanisms are hypothesized to minimize...."

3.) I would rather say the appropriate citation for the "Bayesian brain" hypothesis in line 34 should be Knill and Pouget [2], not Friston [1].

4.) Sentences in lines 27-29 are hard to follow. I would rather say something along the line: "Predictive coding assumes that the brain implements a hierarchical generative model of its sensory inputs, which it optimises by minimising prediction errors between top-down predictions and bottom-up streams of sensory inputs or evidence from lower-level areas" or something along these lines.

5.) Lines 83-85: The increase in the amplitude of the CNV after repeated presentation of rule-congruent or predictable stimuli is really interesting. Could it mean, that this signal is precision-weighted, i.e. increases with an increased certainty (or decreased uncertainty) of the associated predictions?

6.) Lines 130, 133: replace "next" with "following"

7.) Lines 175-180: I assume that the supplementary figure was only placed here for review purposes, as it should not be in the main-text of the published paper (if it is referred to as "SupplX...").

8.) Figures 4 and 5: It would be nice to have x- and y-axis scales not only on the top-left plot, so that the reader does not have to jump back and forth between this and the other panels to find out the exact voltage and time of a specific feature of a specific panel.

9.) Lines 303-308: I'm not sure, what this sentence is supposed to tell me. As this is the final sentence of a discussion paragraph, I kindly ask the authors to split this in a sequence of easier to follow sentences and make abundantly clear what the conclusions for the reader are.

10.) Line 381-383: I'm also not sure what this sentence is supposed to tell me. C.f. above.

Reviewer 2 Report

Review of Ruiz-Martínez, Arjona, & Gómez

“MMN and SPN in paradigms of increasing auditory complexity”

21 February, 2021

This manuscript reports a single EEG study that addresses topics of interest to cognitive neuroscience. I found the paper to be well written and, the experiment, competently executed and analyzed.

The MMN findings are interesting. However, there have been several prior studies that demonstrated an MMN to violation of abstract expectations and, therefore, argued against the habituation/refraction hypothesis. By contrast, the SPN findings are entirely novel and potentially important because they speak to the most critical construct of the predictive coding framework—prediction. There is a weakness, though, in that the authors have not convincingly argued that the putative SPN effects are due to differences in brain activity during the intertrial interval (ITI) rather than during the baseline interval. If participants could predict whether or not the fourth tone would be deviant, a difference in slow-wave amplitude prior to the fourth tone (the baseline period) could have artifactually generated the ITI effect.

Was it the case that after a deviant had been detected the participant could be certain that the next trial would be a standard? If so, expectations just prior to the fourth tone would have been quite different than on trials in which the previous three trials had been standards and a deviant was likely. Further information should be given about the distribution of trials types. Were they random?  Further assurance could be provided by including a figure showing waveforms spanning the four stimuli, with overlays either for standard versus deviant on the current trial or, alternatively, on the preceding trial.

The authors imply some concern over the fact that scalp topographies for the SPN differed from those previously reported. The study by Brunia and van Boxtel (2004) with auditory stimuli is cited. However, the auditory stimuli in that study conveyed performance feedback for a just-emitted motor response. Of greater relevance is the recent study by Bianco and colleagues (2020), which included an auditory condition with no motor task. Encouragingly, the SPN topography in that study was quite similar to the present findings, with positivity distributed over fronto-central sites.

Bianco, V., Perri, R. L., Berchicci, M., Quinzi, F., Spinelli, D., & Di Russo, F. (2020). Modality-specific sensory readiness for upcoming events revealed by slow cortical potentials. Brain Structure and Function225(1), 149-159.

Round 2

Reviewer 1 Report

The authors have nicely addressed my comments.

Reviewer 2 Report

Review of Ruiz-Martínez, Arjona, & Gómez

“MMN and SPN in paradigms of increasing auditory complexity”

8 March, 2021

I’m pleased to say that the revised version of this manuscript fully addresses concerns raised in my previous review. This is a fine paper and one that I expect to be citing.